# Pathway-Directed Therapy in Multiple Myeloma

**DOI:** 10.3390/cancers13071668

**Published:** 2021-04-01

**Authors:** Lukas John, Maria Theresa Krauth, Klaus Podar, Marc-Steffen Raab

**Affiliations:** 1Department of Internal Medicine V, University Hospital Heidelberg, Im Neuenheimer Feld 410, 69120 Heidelberg, Germany; lukas.john@med.uni-heidelberg.de; 2CCU Molecular Hematology/Oncology, German Cancer Research Center (DKFZ), Im Neuenheimer Feld 280, 69120 Heidelberg, Germany; 3Department of Internal Medicine I, Division of Hematology and Hemostaseology, Medical University of Vienna, Währinger Gürtel 18-20, 1090 Vienna, Austria; maria.krauth@meduniwien.ac.at; 4Department of Internal Medicine, Karl Landsteiner University of Health Sciences, Mitterweg 10, 3500 Krems an der Donau, Austria; Klaus.Podar@krems.lknoe.at

**Keywords:** multiple myeloma, signaling pathways, RAS/RAF/MEK/ERK-pathway, PI3K/AKT-pathway, BRAF, mTOR, PIM, p53, c-MYC

## Abstract

**Simple Summary:**

Multiple Myeloma is a cancer of plasma cells in the bone marrow. While effective treatments are available and many patients can now live years with the disease, most patients ultimately run out of treatment options. Pathway-directed therapy looks at genetic aberrations in the tumor cells and tries to blockade the tumor’s Achilles heel. Since myeloma cells show changes in several pathways, which can vary between different patients, agents targeting these pathways often only show activity in few patients. Pathway-directed therapy in myeloma is therefore often combined with personalized medicine, which aims to identify drugs that might interfere with the most important pathways in a particular patient. There are several pathways that can be targeted in myeloma, and, in combination with personalized medicine, some have shown promising results. However, there is still a challenge in identifying suitable patients and preventing resistance to single drugs, most likely caused by other pathways assuming the function of the blockaded one. Further research is therefore required to improve pathway-directed therapy.

**Abstract:**

Multiple Myeloma (MM) is a malignant plasma cell disorder with an unmet medical need, in particular for relapsed and refractory patients. Molecules within deregulated signaling pathways, including the RAS/RAF/MEK/ERK, but also the PI3K/AKT-pathway belong to the most promising evolving therapeutic targets. Rationally derived compounds hold great therapeutic promise to target tumor-specific abnormalities rather than general MM-associated vulnerabilities. This paradigm is probably best depicted by targeting mutated BRAF: while well-tolerated, remarkable responses have been achieved in selected patients by inhibition of BRAFV600E alone or in combination with MEK. Targeting of AKT has also shown promising results in a subset of patients as monotherapy or to resensitize MM-cells to conventional treatment. Approaches to target transcription factors, convergence points of signaling cascades such as p53 or c-MYC, are emerging as yet another exciting strategy for pathway-directed therapy. Informed by our increasing knowledge on the impact of signaling pathways in MM pathophysiology, rationally derived Precision-Medicine trials are ongoing. Their results are likely to once more fundamentally change treatment strategies in MM.

## 1. Introduction

Multiple Myeloma (MM) is a malignant plasma cell disorder affecting approximately 4–6/100,000 people a year. The median age at diagnosis is 66 years [1]. Novel agent-based therapies and earlier initiation of treatment [2] have dramatically increased live expectancy with median progression-free survival (PFS) after first-line currently at approx. 53 months and a 7-year overall survival rate of 62% [3]. Excitingly, recent clinical trials with next-generation novel therapies, namely immunotherapies such as CAR-T cells and bispecific antibodies in particular, have demonstrated impressive PFS and OS even in heavily pretreated patients [4,5,6,7,8,9]. However, most MM-patients ultimately relapse [10]. Despite the advances mentioned above, the prognosis for relapsed and refractory (r/r) patients, in particular when refractory to proteasome inhibitors, immunomodulatory drugs, and anti-CD-38 antibodies, is grim, with a median overall survival of just 8.6 months [11]. MM cells in these r/r patients are often characterized by a high proliferation rate and drug resistance, both to novel agents and conventional chemotherapy. This highlights the need for novel, more effective therapies.

One promising way of finding new therapies is to identify genomic driver events and using specific drugs to target these aberrations. Many tumors are driven or supported by activation of specific kinases and pathways, which, when activated, lead to proliferation of tumor cells [12,13]. Therapies targeting these pathways have yielded impressive results in several solid and hematological malignancies [14,15,16,17,18]. Tumors in which the largest therapeutic success has been achieved are the ones that uniformly share a single driver aberration, such as the bcr-abl-translocation in chronic myeloid leukemia [19]. Unfortunately, the situation in MM is far more complex. MM is a disease with high levels of both inter- and intra-patient heterogeneity [20,21]. Gene expression profiling has shown at least seven subtypes of MM corresponding to genetic lesions widely regarded as the initiating events of tumorigenesis [22]. These lesions consist of recurrent translocations of the IgH-locus on chromosome 14 or gain of additional chromosomes, leading to hyperdiploidy. Translocations bring different oncogenes (such as Cyclin D1 (CCND1), CCND3, fibroblast growth factor receptor 3 (FGFR3), multiple myeloma SET-domain (MMSET/WHSC1), MAF or MAFB) under the control of the IgH enhancer [23]. In the course of the disease, MM-cells acquire additional genetic lesions, such as deletion 17p with subsequent mono-allelic loss of p53, 1q gain with the amplification of 679 genes (i.e., BCL9, PDZK1), and copy-number variations or translocations affecting MYC, as well as somatic mutations of multiple signaling molecules [24,25,26,27]. However, there are only a few mutations that are shared in a significant fraction of patients [28,29]. A large analysis of more than 1200 patients identified a total of 63 driver gene mutations, and only 17 genes showed mutations in more than 5% of patients [26]. This disables the development of a single targeted therapy applicable to all MM-patients [20].

Importantly, besides these genetic abnormalities, myeloma-induced changes of the bone marrow microenvironment result in elevated levels of interleukins, such as IL-6, IGF1, HGF, VEGF, CXCL12, TNFα, BAFF, APRIL, and CCL3 derived from stromal, bone and immune cells, as well as being secreted in a paracrine fashion. This leads to the additional upregulation of signaling cascades, most prominently including the RAS/RAF/MEK/ERK-(also known as MAPK-) pathway, the PI3K/AKT- and the NFκB-pathway, but also the JAK/STAT-, Hedgehog-, Notch-, TGFβ- and the WNT-pathway. Following the milestone approval of imatinib mesylate for CML, pathway-directed therapies have become a major focus of drug development also in MM in order to not only inhibit tumor cell proliferation, survival, migration, and drug resistance, but also to overcome immunoparesis, bone marrow angiogenesis, and bone disease (Figure 1a). While several recent articles excellently review the functional basics of signaling pathways [30], kinases in general [31], as well as STAT3- [32], and NFκB- [33] pathways in MM pathogenesis, we here focus on novel therapeutic strategies that specifically target the RAS/RAF/MEK/ERK- and the PI3K/AKT-pathways and PIM-kinase as well as selected downstream transcription factors. Figure 1 shows an overview of important signaling pathways in MM and drugs targeting those pathways.

## 2. RAS/RAF/MEK/ERK-Pathway Directed Therapies

The RAS/RAF/MEK/ERK-pathway is a pathway of intracellular kinases involved in proliferation, growth, adhesion, and apoptosis. Orthologues of RAS were originally discovered in cancer-causing viruses in rats, leading to the identification of three endogenous human RAS genes, NRAS, KRAS, and HRAS [34,35]. These small GTPases activate RAF-kinases, which in turn phosphorylate MEK and finally ERK-kinases. Mutations in kinases involved in this pathway are found in a variety of human tumors and the pathway and upstream regulators have been successfully targeted in several cancers such as melanoma, hairy cell leukemia, colorectal cancer, and non-small-cell lung cancer [35,36,37,38,39,40].

Mutations of the MAPK-Pathway are among the most common mutations found in MM, with a prevalence of 43–53% of patients [26,28,41,42]. Interestingly the number of patients with mutations seems to be higher in relapsed disease, with up to 72% of patients showing mutations in the r/r setting [41,42,43]. Contradictory results on the prognostic relevance of mutations in the MAPK-pathway have been reported. While some groups found negative effects of NRAS, but not KRAS mutations [44], others observed a negative effect in KRAS but not NRAS mutations [45,46], and others found no prognostic effect at all [47]. While this may in some cases be explained by intra-patient heterogeneity and mutations only present in subclones or at specific sites in the body [21], this controversial finding may also be explained by the fact that, despite the high prevalence, only a few of the mutations seem to actually activate the signaling pathway. In a study by Xu et al. [42], only KRAS G12D and BRAFV600E consistently led to phosphorylation of downstream target ERK. Other mutations were associated with increased pERK-levels only in a small percentage of cases. This suggests, that despite the high prevalence, many of the mutations in the RAS/RAF/MEK/ERK-pathway may not lead to significant dysregulation of this pathway on their own.

RAS itself is very difficult to target, and so far, there is only one specific inhibitor, targeting the KRAS G12C-mutation, which is, however, very rare in MM [48,49]. A study using the farnesyltransferase-inhibitor tipifarnib, which, among other targets, inhibits RAS, but showed limited activity in patients with MM and did not significantly decrease activation of the MAPK-pathway [50].

The focus of MAPK-pathway inhibitors has therefore been on downstream targets of RAS, such as BRAF and MEK. However, there are several pitfalls, and careful patient selection is necessary. For example, it has been shown in other cancers that treatment of patients with BRAF-inhibitors in the presence of nonmutated BRAF can actually lead to a paradoxical activation of the pathway via increased RAS-signaling, in particular when RAS mutations are also present. This is believed to be mediated by a decrease in negative feedback on the RAS-level [51], binding of wild-type BRAF to CRAF, and subsequent MAPK-signaling through CRAF [52,53,54,55].

Care should also be taken on how to identify patients profiting from MAPK-pathway inhibition. A retrospective study by Heuck et al. which examined the effects of treatment with the MEK1/2-inhibitor trametinib nicely illustrates this issue. Despite preselection of patients for known oncogenic mutations in NRAS, KRAS, and BRAF or gene expression profiling suggesting activation of the MAPK-pathway, it was found that only 40% of patients achieved at least a partial response (PR) when trametinib was combined with other substances, and only 10% of patients showed at least a PR when being treated with trametinib as a single agent [56].

This highlights the need for more precise identification of patients profiting from inhibition of the MAPK-pathway. Several recent studies therefore only included patients carrying the BRAF V600E/K mutation, which has been shown to consistently activate ERK and has also been closely examined in a variety of other tumors. In addition, there are several available and potent inhibitors specific for the BRAF V600E/K mutation available, such as vemurafenib, encorafenib, and dabrafenib. All substances have an acceptable safety profile, with the most common side effects being blurred vision, macula edema, cramps, arthralgia, diarrhea, skin rash, decreased left ventricular function, anemia, and thrombocytopenia. The BRAF V600E mutation is also relatively common in MM patients, being present in 2–4% of all newly diagnosed MM-patients with the prevalence of mutations increasing to about 8% in r/r patients and patients with extramedullary disease [42].

Although targeting BRAF is highly effective in tumor types carrying mutant BRAF (i.e., melanoma), rapid resistance against BRAF inhibitors (especially when given as monotherapy) frequently occurs. There are several mechanisms explaining this phenomenon with the most common ones involving the gain of activating mutations up- or downstream of BRAF in either NRAS or MEK, leading to alternative signaling and bypassing of BRAF [57,58,59]. MEK-inhibition has also been shown to induce therapeutic resistance through upregulation of other signaling pathways, such as the PI3K/AKT-pathway [60]. In order to circumvent these mechanisms of resistance, it has been suggested to combine BRAF- with MEK-inhibition to remove two levels of the signaling cascade. Results in melanoma with this approach have been very promising with a dual inhibition being superior to BRAF-inhibition on its own, significantly increasing response rates, PFS, and OS [61]. Current trials in MM therefore also focus on dual inhibition.

First hints that targeting the BRAFV600E mutation in MM-patients can be very effective came from case reports showing promising activity in r/r patients carrying the mutation [62,63,64]. More disappointing results came from basket trials, which included a small number of MM-patients, who did not respond [38,65,66]. However, there are several promising studies recently examining the effects of MAPK-pathway inhibition in MM-patients:

The GMMG-BIRMA-study examined dual inhibition of BRAF and MEK in MM-patients carrying the BRAFV600E or BRAFV600K mutation using a combination of encorafenib and binimetinib. Preliminary results from this study showed an overall response rate (ORR, ≥PR) of 82% with 9 out of 11 patients having at least a PR. The duration of responses in the study was very variable; however, some patients showed responses >1 year, which is very unusual for these heavily pretreated patients included in the study. The BIRMA-study showed that at least for some MM-patients, pathway inhibition is feasible and leads to clinically meaningful responses [67].

Another study, the NCT03091257 trial (“A Study of Dabrafenib and/or Trametinib in Patients With Relapsed and/or Refractory Multiple Myeloma”), tests dabrafenib and/or trametinib in r/r MM-patients. The aim of the study is to examine not only the efficacy of BRAF-inhibition in BRAF-mutated patients but also the effects of MEK-inhibition in patients who only present with RAS mutations.

The NCT03312530 study evaluated the safety and efficacy of the MEK-inhibitor cobimetinib in combination with venetoclax and/or atezolizumab in patients with r/r MM. The study also included patients without any mutations in the MAPK-pathway. Cobimetinib showed no activity on its own, but some patients receiving a combination therapy with venetoclax or with venetoclax and atezolizumab showed a response with an ORR (≥PR) of 27% and 29%. While some of this effect may be due to the presence of a translocation t(11;14) in several patients, there were also patients without t(11;14) responding to the combination, in some cases even with durable responses [68].

The NCT02407509-Phase I trial examined the safety and efficacy of the experimental pan-RAF-inhibitor CH5126766 (also known as VS-6766, and previously named RO5126766) in patients with RAS/RAF/MEK pathway mutations. The study included patients with solid tumors but also seven MM patients, one of whom, while carrying a G12V-mutation, showed a PR. The only patient carrying a BRAFV600E mutation, however, did not respond [66].

The MyDRUG-trial (ClinicalTrials.gov Identifier: NCT03732703) is a molecularly stratified umbrella trial examining, among other combinations, a combination of the MEK-inhibitor cobimetinib with ixazomib and pomalidomide in patients with MAPK mutations.

In addition, there are several basket-trials that also include MM patients such as the TAPUR (ClinicalTrials.gov Identifier: NCT02693535) and CAPTUR (ClinicalTrials.gov Identifier: NCT03297606) trials. However, no results have been published so far.

In summary, only a few MM patients respond to inhibition of the RAS/RAF/MEK/ERK-pathway; however, inhibition of the MAPK-pathway shows promising activity in a subset of patients. Inhibition seems particularly feasible in patients with the BRAF V600E mutation, with the BIRMA study showing high efficacy and no unexpected side effects in a small cohort of r/r MM-patients. Further studies are needed to elucidate if there are other predictors for effective MAPK-pathway inhibition in MM, such as selecting patients by actual activation markers of key pathway regulators using phospho-immunohistochemistry or gene-set enrichment analysis of downstream regulatory networks.

## 3. PI3K/AKT-Pathway-Directed Therapies

### 3.1. AKT

Protein kinase B (AKT) is a key serine/threonine-kinase within the PI3K/AKT/mTOR-pathway. AKT is activated by phosphoinositide-3-kinase and localized to the plasma membrane, where it activates several downstream targets involved in proliferation, cell survival, plasma cell development, and angiogenesis, such as mammalian target of rapamycine (mTOR), MDM2, GSK3beta, FKHR, IkK, FoxO, and PRAS40 [69,70].

The rationale for AKT-inhibition in MM is based on high levels of activation in MM cells when compared to cells from patients with MGUS or smoldering MM and inhibition leading to decreased viability in cell lines [71]. Interestingly mutations in the AKT-pathway are not commonly found in MM [72], suggesting an alternative mechanism of AKT-pathway activation as a consequence of other deregulated pathways, such as the MAPK pathway, Il-6 signaling, or the NFκB network [73,74,75,76,77].

Trials testing the AKT-inhibitor perifosine, an alkyl-phospholipid-analogue-targeting signaling pathways at the cell membrane, among them the AKT-pathway, showed mixed results. Perifosine showed limited efficacy on its own or in combination with dexamethasone, with only 13% of r/r patients achieving at least a PR [78]. In a phase I/II combination trial with lenalidomide and dexamethasone, the response rate (≥PR) was 50%; however, most patients in this study were only refractory to thalidomide, and only few patients had received lenalidomide before. Therefore, the effect cannot be only attributed to perifosine and might be at least partly due to lenalidomide [79]. Another phase I/II-trial testing perifosine in combination with bortezomib and dexamethasone showed a response rate (≥minor remission (MR)) of 41% in patients relapsed or refractory to bortezomib, however, with a PFS of only 6.8 months [80]. A phase III trial (ClinicalTrials.gov Identifier: NCT01002248) further evaluating this combination was stopped at the first interim analysis due to a lack of benefit in responses (ORR ≥ PR 20% vs. 27 %) and PFS (22.7 weeks (95% CI 16.0–45.4) in the perifosine arm and 39.0 weeks (18.3–50.1) in the placebo arm) [81].

Novel, more specific AKT-inhibitors like the small-molecule inhibitor afuresertib are currently being tested and show similar results as perifosine, with a single-agent response rate (≥PR) of 8.8% and a median duration of response of 319 days [82]. We and others hypothesize that this small group of treatment-responsive patients likely belongs to the AKT-driven subset of MM patients who benefit most from AKT inhibition [83].

Afuresertib has also been investigated in a phase I/II-trial (ClinicalTrials.gov Identifier: NCT01445587) in combination with bortezomib and preliminary data showed a response rate (≥PR) of 41% in r/r MM-patients [84].

It was also shown that RAS and AKT constitute independent driver pathways, suggesting a rationale for combined MAPK-pathway and AKT-pathway inhibition [85,86,87]. However, a trial examining the combination of MEK and AKT-inhibition using trametinib and afuresertib was discontinued due to poor tolerability, in particular, high rates of diarrhea and skin disorders [88]. There is another trial testing trametinib in combination with GSK2141795, another AKT-inhibitor with results still pending (ClinicalTrials.gov Identifier: NCT01989598). Since the side effects reported by Tolcher et al. [88] are considered class effects of both MEK and AKT-inhibitors and caused by on-target effects in healthy tissue, combination therapies might generally be too toxic to work.

The phase II MATCH-trial (ClinicalTrials.gov Identifier: NCT02465060) also stratifies patients carrying AKT-mutations to receive the novel, mutation-targeting AKT-inhibitor capivasertib; however, no results are available yet.

### 3.2. MTOR

MTOR is a downstream target of AKT and plays a crucial role in MM-cell proliferation and protein synthesis. While preclinical models showed promising results [89], clinical studies only showed very limited activity of the mTOR-inhibitors everolimus and temsirolimus [90,91]. Combination with bortezomib and lenalidomide led to better, albeit still low, response rates (≥PR) of up to 33% [92,93]. Limited efficacy was also observed with the TORC1/TORC2 inhibitor sapanisertib (TAK-228), with a minimal response in only one out of 33 MM-patients after single-agent therapy [94].

In summary, response rates of AKT-inhibitors, even when given in combination with other agents, are generally low. Reasons are unknown and under investigation. While AKT is overexpressed in many MM-patients, some in vitro studies showed that MM-cells fall into two subgroups, AKT-dependent and independent [95]. Therefore, the identification of activation markers of the AKT pathway may be predictive of the response to respective inhibitors. On the other hand, the PI3K/AKT/MTOR-pathway is very large and intertwined with other pathways, so inhibition of an upstream target such as AKT or mTOR might not be enough to inhibit downstream effectors [77].

## 4. PIM-Directed Therapies

There are three isoforms of the proviral insertion site of Moloney murine leukemia virus (PIM)-kinases, small serine/threonine kinases. PIM-1 was originally described in a lymphoma-causing murine virus [96], and members of the PIM-family have been found to be overexpressed in several hematological malignancies [97,98]. PIM-expression has been shown to contribute to proliferation, survival, cell cycle dysregulation, and bone destruction, as well as chemoresistance in some tumor models, and MM shows particularly high expression of PIM-2 [99,100,101,102,103,104]. Interestingly, PIM-kinases seem to exert part of their effects by activating other oncogenic pathways such as the mTOR and MYC-pathway [97,99,101,105].

There are several PIM-kinase inhibitors available; however, only LGH447/PIM447, a pan-PIM-kinase inhibitor, has so far reported clinical results. In a phase I study with heavily pretreated r/r MM patients, LGH447 was well tolerated with mostly hematological adverse events comparable to proteasome inhibitor or Imid-based therapy. Surprisingly, for a single-agent phase I trial, LGH447 showed a disease control rate of 72.2% and a PFS of 10.9 months [105].

Inhibitors of the PIM-kinase have also been combined with inhibitors of the RAS/RAF/MEK/MAPK- and PI3K/AKT/mTOR- pathway and in combination with pomalidomide and dexamethasone [106]. However, a phase I/II study (ClinicalTrials.gov Identifier: NCT02144038) examining the effect of a combination of a LGH447/PI3K-inhibitors was discontinued due to toxicity. Taken together, while PIM-inhibition showed promising activity in a phase I study, there are currently no clinical trials examining this further. While there is preclinical rationale for combining PIM-inhibition with pomalidomide and dexamethasone [107], which leads to downregulation of IRF4 and convergently inhibits protein translation through inhibition of mTORC1 and c-MYC, there are so far no studies examining combination therapies as further development of this compound has been abandoned.

## 5. Transcription Factor-Directed Therapies

As terminal effectors of signaling cascades, transcription factors (TFs) coordinate cell differentiation, proliferation, survival, and migration. Direct modifications, epigenetic changes, and extrinsic or intrinsic activation or inhibition of oncogenic or suppressor TFs, respectively, are responsible for deregulated transcriptional programs in solid and hematologic malignancies, including MM. In addition, TFs are associated with oncogenic addiction, the dependency on prolonged oncogene activity. While nuclear hormone receptor (NHR)-TFs (e.g., the estrogen, progesterone, and steroid receptors) belong to the most common therapeutic targets in cancer (e.g., in breast and prostate carcinoma, lymphoid malignancies, and MM, respectively), non-NHR-TFs (among them p53 and c-MYC) have been widely considered as “undruggable”. Nevertheless, this paradigm has been revised. Indeed, approaches to therapeutically target TFs are rapidly evolving and among today’s most promising anti-tumor strategies. These strategies include inhibition of their expression, induction of their degradation, disruption of their interactions with critical binding partners, or epigenetic modulation of their chromatin accessibility [108].

### 5.1. C-MYC Directed Therapies

The MYC family of TFs controls a wide range of cellular processes, such as proliferation, transcription, translation, metabolism, and apoptosis [109]. Its role in oncogenesis was first identified as a homolog of an avian retrovirus and as being overexpressed in Burkitt’s lymphoma due to the translocation t(8;14) [110,111]. In MM, c-MYC is a downstream target of the RAS/RAF/MEK/MAPK- and the PI3K/AKT/mTOR-pathways. It can therefore be considered as a master regulator and integrator of tumor-associated cellular signaling networks [41,97,101,112,113,114,115]. Overexpression of c-MYC in germinal center B-cells leads to the development of MM in a mouse model [116], while enhanced activation of c-MYC has been observed in up to 67% of newly diagnosed MM-patients as opposed to MGUS [113]. Genetic events affecting the MYC-pathway are associated with disease progression and aggressiveness [24,26].

C-MYC is transcriptionally regulated by the Bromodomain and Extra-Terminal-(BET) subfamily of bromodomain proteins, and BET-inhibition induces a potent antiproliferative effect both in vitro and in vivo [117]. Nevertheless, in a phase I study with the BET- inhibitor OTX015, none of the MM patients responded [118]. Preliminary data of a phase I study using the BET-inhibitor RO6870810 in combination with daratumumab showed an ORR of 16.7%, with some daratumumab-refractory patients also responding [119]. Another phase I study is currently evaluating the BET- inhibitor CPI-0610 in patients with previously treated multiple MM (ClinicalTrials.gov Identifier: NCT02157636) [120]. However, results from clinical studies targeting BET-proteins in MM are so far disappointing. A possible explanation for this might be a consequence of other frequently upregulated pathways in MM, such as the WNT/β-catenin-pathway, the RAS/RAF/MEK/ERK-pathway, and the NFκB-pathway and upregulation of the anti-apoptotic proteins MCL-1 and BCL2, which have all been shown to contribute to resistance to BET-inhibition in other cancers [121,122,123,124,125,126,127]. An alternative to BET-inhibition might be to target c-MYC directly. Direct c-MYC inhibitors include siRNA DCR-MYC, an MYC-directed LNP-formulated siRNA, the MYC-MAX dimerization inhibitors 10058-F4, 10074-G5, as well as the translation inhibitors CMLD010509/rocaglate and TGR-1202 [128,129,130,131,132]. Another interesting novel strategy showing promising in vitro efficacy is protein targeting chimeric molecules (PROTACs) targeting BET-family members [133].

### 5.2. p53 Directed Therapies

The transcription factor p53 is known as a gatekeeper for cell cycle arrest and apoptosis. Germline mutations lead to Li-Fraumeni-syndrome [134], and loss of the p53 allele or disabling p53 mutations are among the most frequent mutations in human cancer and lead to dismal outcomes [135,136]. One of the most common high-risk genetic aberrations in MM involves deletion of the p53-locus on chromosome 17p. While only present in 8% of newly diagnosed MM-patients, the percentage of loss or mutation of p53 increases up to 45% in the r/r setting [137]. Deletion of 17p leads to a dismal outcome in MM-patients and “double-hit” myelomas, with a concurrent mutation in the other allele having a particularly bad prognosis, suggesting a high relevance of the p53-pathway in MM [25,27].

Nutlins increase the activity of p53 by inhibiting its interaction with MDM2, an E3-ubiquitin-protein ligase, thereby preventing its degradation [138]. Potent in vitro anti-myeloma activity was induced by nutlin-3, and synergistic effects with melphalan and bortezomib were observed [139,140,141]. However, this mechanism is dependent on a preserved p53-pathway and the presence of wild-type p53 [139,142,143]. Idasanutlin is currently being evaluated in combination with ixazomib and dexamethasone in a phase I/II trial in r/r MM-patients with deletion 17p (ClinicalTrials.gov Identifier: NCT02633059). Furthermore, the phase I NCT03031730-trial investigates the effects of the MDM2-inhibitor KRT-232 (AMG232) in addition to carfilzomib/lenalidomide/dexamethasone in r/r-patients.

RITA, another inhibitor of the p53/MDM-2 interaction [144], demonstrated significant anti-myeloma activity, independent of the p53 status and even in tumor cells resistant to nutlin-3. Consequently, the combination of nutlin with RITA triggered synergistic cell killing [142,143].

PRIMA-1 is a small-molecule compound specifically designed to restore activity of mutant p53 with antitumoral activity in myeloma cell lines [145,146,147,148].

Since its coding gene is located in close proximity to the p53 locus on chromosome 17p co-deletion of p53 and POLR2A, the largest subunit of RNA polymerase II has been proposed as a collateral vulnerability target. The antibody-drug conjugate HDP-101 couples a synthetic version of amanitin, targeting POLR2A with a BCMA-antibody. HDP-101 has shown promising in vitro activity and in vivo tolerability in MM, and a phase I/II study will be initiated in early 2021 [149,150].

### 5.3. Other TF-Directed Therapies

Besides c-MYC and p53, other signal-activated TFs in myeloma include members of the NFκB- and the STAT3 family, as well as AP-1, but also SP-1, GFI-1, C/EBPβ, E2F1, HIF-1α, and PU.1. Preclinical but also phase I–II clinical trials targeting these TFs are ongoing [108].

## 6. Conclusions

Management of MM, even in the era of novel agents, remains challenging, as the majority of patients develop resistance to established forms of treatment over time and relapse. Despite promising results and lasting remissions in some patients with MM, the overall efficacy of pathway-directed therapies observed so far has been modest [9]. Potential explanations for the lack of high response rates upon single-pathway inhibition include inherent challenges in myeloma such as high mutational load, spatial and temporal heterogeneity, as well as the absence of unifying driver events. Ongoing studies in myeloma aim to further increase our knowledge on deregulated pathways, the identification of predictive markers to select probable responders, the optimization of timing and/or sequencing of drug administration, and mechanisms that lead to specific pathway inhibitor resistance (i.e., redundant pathways). Moreover, additional efforts are aiming to define the ability of pathway inhibitors to re-sensitize tumor cells to conventional myeloma drugs such as proteasome inhibitors or Imids, naked antibodies, and future next-generation immunotherapies, CAR T cells and BiTEs in particular [119]. Indeed, rationally derived Precision Medicine trials in multiple myeloma with pathway-directed therapies alone or in combination with conventional or novel therapies are already ongoing and include the BIRMA, MATCH, TAPUR, CAPTUR, and the MyDrug trial, for which results are eagerly awaited. An overview of published and ongoing trials investigating pathway-directed therapy is shown in Table 1.

## Figures and Tables

**Figure 1 cancers-13-01668-f001:**
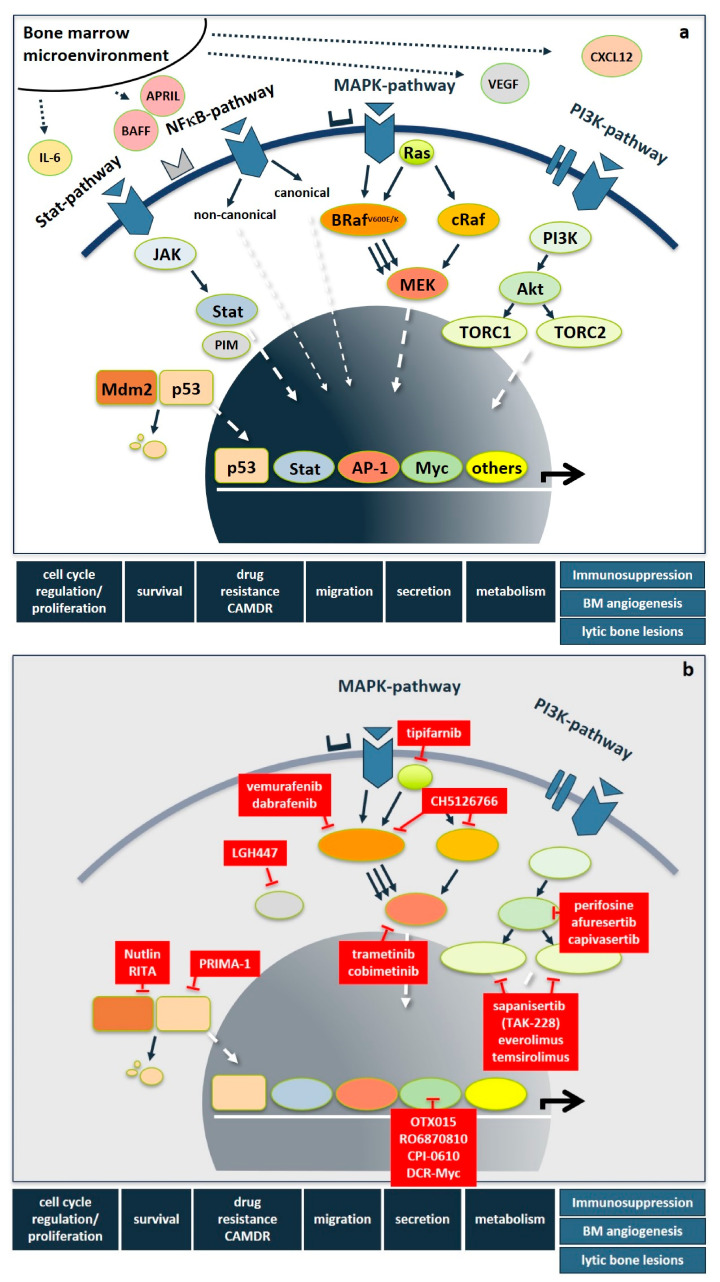
Deregulated signaling cascades and pathway-directed therapies in multiple myeloma. (**a**) Signaling pathways in multiple myeloma. In multiple myeloma, genetic abnormalities and bone marrow microenvironment-driven deregulations, such as increased levels of IL-6, IGF1, HGF, VEGF, CXCL12, TNFα, BAFF, APRIL, and CCL3 derived from stromal, bone, and immune cells result in the activation of multiple signaling pathways, most prominently including the RAS/RAF/MEK/ERK-, PI3K/AKT-, NFκB-, and STAT-pathway but also the WNT-, Hedgehog-, and TNFα-pathway. They trigger tumor cell proliferation, survival, drug resistance, migration, secretion of humoral factors but also promote immunoparesis, bone marrow angiogenesis, and bone disease. (**b**) Pathway-directed therapies. Abbreviations: Bone marrow (BM), cell adhesion-mediated drug resistance (CAMDR), interleukin-6 (IL-6), B-cell activating factor (BAFF), A proliferation inducing ligand (APRIL) vascular endothelial growth factor (VEGF), C-X-C motif chemokine 12 (CXCL12), Signal transducer and activator of transcription (STAT), nuclear factor kappa B (NFκB), phosphoinositide 3-kinase (PI3K), Janus kinase (JAK), Mouse double minute 2 homolog (Mdm2), CREB-regulated transcription coactivator 1 (TORC1), CREB-regulated transcription coactivator 1 (TORC2), protein kinase B (AKT), RAS-kinase (RAS), B-RAF-kinase (BRAF), C-RAF-kinase (cRAF), mitogen activated protein kinase (MAPK), mitogen activated protein kinase kinase (MEK), proviral insertion site of Moloney murine leukemia virus kinase (PIM), Activator protein-1 (AP-1).

**Table 1 cancers-13-01668-t001:** Overview on pathway directed therapies in multiple myeloma.

Study	Drugs	Study Type	Efficacy	Patient Selection
RAS/RAF/MEK/ERK-pathway				
Alsina et al. [50]	Tipifarnib	Phase II	64% stable disease, 0% ≥ PR	r/r MM
Heuck et al. [56]	Trametinib	Retrospective cohort	10% ≥ PR	Mutations in NRAS, KRAS, BRAF, MAPK-activation in GEP
Hyman D.M. et al. [38]	Vemurafenib	Phase II Basket-trial	No responses in the 5 MM patients	BRAFV600 mutated
NCI-MATCH [65]	Dabrafenib + Trametinib	Phase II Basket-trial	No response in myeloma patients	BRAF V600E/R/K/D mutated
BIRMA-Study [67]	Encorafenib + Binimetinib	Phase II	ORR (≥PR) 82%	BRAFV600-mutated
NCT03091257	Dabrafenib and/or Trametinib	Phase I	Ongoing	BRAF/KRAS/NRAS mutated
NCT03312530 [68]	Cobimetinib + Venetoclax ± Atezolizumab	Phase I/II	ORR (≥PR) 27%/29% in the combination arms	-
Guo et al. [66]	CH5126766 (VS-6766/ RO5126766)	Phase I	PR in 1/7 myeloma patients	Solid tumors and myeloma with RAS/RAF/MEK pathway mutations
MyDRUG-trial (NCT03732703)	Cobimetinib + Dexamethasone + Ixazomib/Pomalidomide	Phase I/II Umbrella-trial	Ongoing	RAF/RAS-mutation
TAPUR (NCT02693535)	Vemurafenib + Cobimetinib	Phase II Basket-trial	Ongoing	BRAFV600 E/D/K/R mutated
CAPTUR (NCT03297606)	Vemurafenib + Cobimetinib	Phase II < Basket-trial	Ongoing	BRAF V600 mutated
AKT-pathway				
Richardson et al. [78]	Perifosine (+ Dexamethasone)	Phase II	38% PR + MR after addition of dexamethasone	r/r MM
Jakubowiak et al. [79]	Perifosine + Lenalidomide + Dexamethasone	Phase I	ORR (≥PR) 50%	r/r MM, no previous therapy with lenalidomide required
Richardson et al. [80]	Perifosine + Bortezomib + Dexamethasone	Phase I/II	ORR (≥MR) 41%, 32% in bortezomib-refractory patients	r/r MM
Richardson et al. [81]	Perifosine + Bortezomib + Dexamethasone	Phase III	ORR (≥PR) 20% vs. 27% in the placebo arm)	Phase III
Spencer et al. [82]	Afuresertib	Phase I	ORR (≥PR) 8%, long median PFS in responding patients (319 days)	r/r MM
Voorhees et al. [84]	Afuresertib + Bortezomib + Dexamethasone	Phase I/II	Preliminary data: ORR (≥PR) 41% in phase I part	r/r MM
Tolcher et al. [88]	Trametinib + Afuresertib	Phase I/II	Discontinued due to toxicity	r/r MM, relapsed triple negative breast or endometrial cancer
NCT01989598	GSK2141795 + Trametinib	Phase II	Ongoing	r/r MM
NCI MATCH	Capivasertib	Phase II	Ongoing	AKT-mutated
Günther et al. [90]	Everolimus	Phase I	ORR (≥PR) 7% (1/15, maximum PFS 3 (months)	r/r MM
Farag et al. [91]	Temsirolimus	Phase II	ORR (≥PR) 6% (1/16)	r/r MM
Yee et al. [93]	Everolimus + Lenalidomide	Phase I	ORR (≥MR) 65%	r/r MM, no previous lenalidomide required
Ghobrial et al. [92]	Temsirolimus + Bortezomib	Phase I/II	≥PR 33%	r/r MM, no previous bortezomib required
Ghobrial et al. [94]	Sapanisertip (TAK228)	Phase I	1/31 myeloma patients with MR	r/r MM
PIM-kinase pathway				
Raab et al. [105]	PIM 447(LGH447)	Phase I	ORR (≥PR) 9%, disease control rate 72%, median PFS 10.9 months	r/r MM
NCT02144038	PIM 447(LGH447) + BYL719	Phase I/II	Discontinued due to toxicity	r/r MM
c-MYC pathway				
Amorim et al. [118]	OTX015	Phase I	No activity in the myeloma group	r/r MM, lymphoma
NCT02157636	CPI-0610	Phase I	Ongoing	r/r MM
NCT03068351 [119]	RO6870810 + Daratumumab	Phase I	ORR (≥PR) 16.7%	r/r MM, no previous daratumumab required
Tolcher et al. [128]	DCR-MYC	Phase I	No published results available	r/r MM, advanced solid tumors, lymphoma
p53 pathway				
NCT02633059	Idasanutlin + Ixazomib + Dexamethasone	Phase I/II	No published results available	r/r MM with del17p
NCT03031730	KRT-232 (AMG232) + Carfilzomib + Lenalidomide + Dexamethasone	Phase I	ongoing	r/r MM
Strassz et al. [150]	HDP-101	Phase I/II	Due to start Q1/2021	r/r MM

Abbreviations: MM: Multiple Myeloma, ORR: overall response rate, PR: partial remission, MR: minor remission.

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
