# Peer review of "Pathway-Directed Therapy in Multiple Myeloma"

_cancers, 2021, doi:10.3390/cancers13071668_

Round 1

Reviewer 1 Report

The authors reviewed pathomechanisms of myeloma patients and proposed potential pathway-driven therapeutic strageties. Though interesting, There are still some questions for this manuscript to be published.

  1. Though pathway-directed therapy interesting and attracted, current results are not good enough to suppport it. Such as in MAPK pathway section, only small cases have been reported and reviewed. The same situations can also been seen in other pathway sections. The authors should put more data or discussion about current difficulties.
  2. The authors should have more discussion about the pathomechanisms of different pathways in myeloma patients in order to make pathway-directed therapy reasonable.
  3. The figure 1 is not clear and have some defects, especially in figures 1b.

Reviewer 2 Report

Revision

Pathway-directed therapy in Multiple Myeloma by

Lukas John, Maria Theresa Krauth, Klaus Podar, Marc-Steffen Raab

  • Well written; important review on precision medicine targeting signaling pathways alterations in MM. Interesting and updated review on a very interesting theme.
  • Minor revisions asked:
  • In line 74: “…. prominently including the RAS/RAF/MEK/ERK-(also MAPK-) …” it is not clear. Should be re phrase by (known as MAPK).
  • In fig 1 legend, “bone marrow microenvironment-driven deregulations” is not clear in the figure. Please make it clearer in the figure and address it more in the text.  Even that it is not the main focus of this review, some information on this should be included.
  • In fig 1b the name of the molecules are not visible. Layout of the figure may be improved.
  • Line 99: “tumor causing virus in mice” – correct
  • Line 103: give examples of tumors.
  • Toxicities occurred in the trials should be discussed not only enumerated.
  • Line 288: Phase I trials are not meant to evaluate efficacy. Please re write the idea. Explain toxicity.
  • Clarify the reason for combining PIM-inhibition with pomalidomide and dexamethasone
  • Line 296: add a comma after “As terminal effectors of signaling cascades”
  • Give examples of nuclear hormone receptor (NHR)-TFs commonly used to treat cancer/MM.
  • I would like to see a discussion or hypothesis explaining the controversial, or even poor results, of BET- inhibitors in MM.

Round 2

Reviewer 1 Report

The author responsed and modified the manuscript according to reviews' comments. Therefore, I think it is suitable for publish in the journal.